# Model Identifies Genetic Predisposition of Alzheimer’s Disease as Key Decider in Cell Susceptibility to Stress

**DOI:** 10.3390/ijms222112001

**Published:** 2021-11-05

**Authors:** Ioanna C. Stefani, François-Xavier Blaudin de Thé, Cleo Kontoravdi, Karen M. Polizzi

**Affiliations:** 1Department of Chemical Engineering, Imperial College London, London SW7 2AZ, UK; ioanna.stefani06@gmail.com (I.C.S.); francoisxavier.dethe@gmail.com (F.-X.B.d.T.); 2Imperial College Centre for Synthetic Biology, Imperial College London, London SW7 2AZ, UK

**Keywords:** unfolded protein response, amyloid precursor protein, beta-amyloid, mathematical modelling, endoplasmic reticulum stress, neurodegeneration

## Abstract

Accumulation of unfolded/misfolded proteins in neuronal cells perturbs endoplasmic reticulum homeostasis, triggering a stress cascade called unfolded protein response (UPR), markers of which are upregulated in Alzheimer’s disease (AD) brain specimens. We measured the UPR dynamic response in three human neuroblastoma cell lines overexpressing the wild-type and two familial AD (FAD)-associated mutant forms of amyloid precursor protein (APP), the Swedish and Swedish-Indiana mutations, using gene expression analysis. The results reveal a differential response to subsequent environmental stress depending on the genetic background, with cells overexpressing the Swedish variant of APP exhibiting the highest global response. We further developed a dynamic mathematical model of the UPR that describes the activation of the three branches of this stress response in response to unfolded protein accumulation. Model-based analysis of the experimental data suggests that the mutant cell lines experienced a higher protein load and subsequent magnitude of transcriptional activation compared to the cells overexpressing wild-type APP, pointing to higher susceptibility of mutation-carrying cells to stress. The model was then used to understand the effect of therapeutic agents salubrinal, lithium, and valproate on signalling through different UPR branches. This study proposes a novel integrated platform to support the development of therapeutics for AD.

## 1. Introduction

Alzheimer’s disease (AD) is a progressive neurodegenerative disorder characterized by the accumulation and aggregation of misfolded protein structures: extracellular beta-amyloid (Aβ) in senile plaques and intracellular tau aggregates in neurofibrillary tangles [1]. The presence of these structures interferes with several cell signalling mechanisms, causing cellular toxicity and inducing endoplasmic reticulum (ER) stress [2,3,4]. The ER stress response triggers an intracellular mechanism called the unfolded protein response (UPR), which acts as a feedback loop to restore homeostasis. However, when stress persists, cells commit to apoptosis. UPR activation has been associated with a number of different diseases, including neurodegeneration, diabetes, and cancer [5]. Increased expression of UPR markers—including the 78 kDa glucose-regulated chaperone GRP78/BiP and the active phosphorylated form of PERK, were demonstrated to be present in the temporal cortex and hippocampus in AD human brain specimens by Hoozemans et al. [4]. Recent studies on in vivo mouse models have highlighted the involvement of the UPR in neurodegeneration, making it a possible treatment avenue [6,7]. Notably, it has been shown that manipulation of the UPR can induce cellular adaptation and improvement of the disease phenotype [8,9,10,11]. Accumulation of misfolded proteins is recognized by three ER transmembrane sensors: protein kinase R-like ER kinase (PERK), inositol requiring enzyme 1 (IRE1), and activating transcription factor 6 (ATF6) (Figure 1). Activation of the PERK pathway results in the phosphorylation of eukaryotic initiation factor (p-eIF2α), which both represses protein synthesis [12,13] and results in the overexpression of activating transcription factor 4 (ATF4), in turn triggering the expression of the pro-apoptotic factor C/EBP-homologous protein (CHOP). IRE1 signalling is the most conserved pathway and a key element responsible for the duration as well as the magnitude of the UPR [14,15]. When activated, it leads to the unconventional splicing of a small intron from the X-box binding protein 1 (XBP1), forming the XBP1 spliced (XBP1s) mRNA [16]. The resulting protein is an active transcription factor (TF), which upregulates the genes associated with the ER-associated degradation (ERAD) pathway as well as chaperones and foldases. Once activated, ATF6 is translocated to the Golgi apparatus, where it is cleaved by resident proteases, releasing its active form, a 50 kDa TF called p50. This protein induces the expression of XBP1, along with ER chaperones [17]. PERK and ATF6 are believed to be activated before IRE1, promoting mainly ER adaptation, while IRE1 induces both survival and pro-apoptotic signals [18].

The UPR is a highly complex system that involves many pathways, characterised by protein and gene interactions, as well as biophysical processes [11]. Relevant experimental data to elucidate the role of the UPR in AD in humans are difficult to obtain due to the late stage at diagnosis and the length of time it takes for disease to develop. In vitro and animal models of the disease allow us to interrogate the pathways to a larger extent, but abstraction to the human body is difficult. A first step towards that could involve the use of mathematical models to enable investigation of individual pathways and help relate different biological and transport phenomena. Preliminary models based on data from in vitro or animal model systems can be used to compare the role of different mutations, identify promising points of intervention, and investigate treatment avenues. In this study, we established a proof-of-concept system by creating cell lines that mimic a genetic predisposition to Alzheimer’s, subjecting them to environmental stress, and using the resulting experimental data to formulate a mathematical model that can be used to explore the effects of different treatments in silico. Specifically, we developed mutant cell lines overexpressing the wild-type amyloid precursor protein (APP_WT_) and two APP mutants: Swedish (APP_S_) and Swedish-Indiana (APP_S-I_, collectively called APP_MUT_) APP695. These two mutations are associated with familial AD (FAD), a form of AD that is linked to genetic predisposition and is responsible for most cases of early disease onset. Using cell lines allowed us to characterise the short-term response to stress and create a mathematical model that helps to rationalise the experimental observations. The results suggest that mutations confer a different response to stress, which is accurately captured within the model.

## 2. Results

### 2.1. Mutations in APP Associated with FAD Lead to Different Magnitude of ER Stress Response Signalling

Most of the early onset AD cases are linked to a genetic predisposition. One example is mutations in the amyloid precursor protein (APP) gene, e.g., missense mutations in codons 670 and 671 (Swedish mutation) or in codon 717 (Indiana mutation). Swedish and Indiana mutations affect both the processing of APP by γ-secretase and the catabolic pathways responsible for the degradation of these fragments [19,20,21], resulting in an increase in Aβ_42_ concentration or the ratio of Aβ_42_/Aβ_40_ [21,22,23], respectively. To establish a model system of FAD-associated genetic predisposition, three human SK-N-SH neuroblastoma cell lines were created, overexpressing APP_WT_ and the Swedish and Swedish-Indiana APP695 mutations (APP_S_ and APP_S-I_, respectively). Overexpression of APP695 protein in the three transfected cell lines was confirmed by qRT-PCR (data not shown) and Western blotting (Figure 2). To mimic additional environmental stress, cells were treated with 1 µg/mL of tunicamycin (Tm), a well-characterized ER stress inducer that inhibits protein glycosylation, leading to a build-up of unfolded proteins in the ER [2,24]. mRNA was isolated at different time points over the course of 8 h, and gene expression analysis was carried out using qRT-PCR. The mRNA from the UPR responsive genes BiP, ATF4, total XBP1 (XBP1T), and CHOP was upregulated with Tm treatment, as was the amount of XBP1 splicing, indicating that the stress response was activated (Figure 3a–e). Metabolic activity as assessed by the MTS assay was reduced at 6 h post Tm addition (Figure 4a), rebounded at 8 h, but was again decreased at 24 h. This temporary slowdown in metabolism could have been due to G1 arrest of cells induced by tunicamycin [25]. Culture viability was over 90% for all cell lines at all time points assessed (Figure 4b), although at 24 h post tunicamycin addition, there was a statistically significant decrease in viable cell density (Table 1).

Our gene expression analysis indicated activation of UPR signalling in response to environmental stress, at least in part, for all cell lines. The most strongly induced UPR marker was CHOP mRNA (Figure 3a), which was upregulated in all cell lines at all time points assessed, ranging from ~11-fold increase in APP_WT_ to nearly 70-fold in APPs. CHOP is a pro-apoptotic factor whose expression is regulated by the transcription factor ATF4, the levels of which increase upon the activation of PERK.

Splicing of XBP1 indicated activation of IRE1 signalling (Figure 3b) [26]. APP_S_ had statistically elevated levels of spliced XBP1 at all time points, with strong activation at 6 and 8 h post tunicamycin addition. APP_S-I_ had similar increases in the average fold-change; however, due to high variations in the data, only the 3-h samples were statistically significant. In APP_WT_, the level of spliced XBP1 mRNA was low and remained so throughout the experiment. 

ATF6 signalling is activated by translocation of the membrane protein to the Golgi apparatus, where it undergoes proteolysis, releasing an active transcription factor [27]. We attempted to monitor this reaction using Western blotting, but levels of ATF6 were too low to detect (data not shown). However, ATF6 activation eventually leads to increased transcription of XBP1 further downstream [26]. Hence, we used the total expression levels of this gene as a marker for ATF6 activation (Figure 3c). XBP1 levels were elevated in APP_S_, particularly at 6 and 8 h post tunicamycin addition, but remained relatively low and constant in APP_WT_. For APP_S-I_, there was an upregulation of total XBP1 levels beginning at 4 h through to the end of the experiment.

Finally, we measured the levels of mRNA for two UPR markers that are not connected to a single pathway. BiP is a chaperone that is a major component of ER stress signalling [28]. Once again, upregulation was the strongest in APPs, with statistically significant upregulation at all time-points and reaching a maximum of 25-fold increase over time zero at 8 h (Figure 3d). For APP_S-I_, upregulation began at 4 h post tunicamycin addition and did not reach the same high levels as for APPs (maximum of 16-fold upregulation at 8 h). Interestingly, while BiP levels increased steadily across time in APP_MUT_, the activation profile in wild-type showed a low response until 8 h. BiP is regulated by the ATF6 or IRE1 signalling pathways; thus, its presence suggests activation of either or both of these in all cell lines. 

ATF4 is regulated on both a transcriptional and translational level and its overexpression leads to an increase in CHOP expression. However, it is not currently known which signalling branch is responsible for ATF4 transcriptional control [29]. In our analysis, there was a nearly significant elevation in APP_WT_ beginning at 3 h (Figure 3e, *p* = 0.10), and the activation was sustained at a similar level throughout the experiment (approximately 3–4-fold upregulation, 0.07 < *p* < 0.1). On the other hand, in both APP_S_ and APP_S-I_, ATF4 upregulation was weaker at early time points but ultimately increased to a higher degree than APP_WT_ at 8 h post tunicamycin addition.

Taken together, the results from the qRT-PCR analysis suggest that APPs is the most stressed cell line and APP_WT_ the least, with APP_S-I_ showing intermediate stress levels. The fact that APP_MUT_ is more stressed than APP_WT_ suggests that the higher accumulation of Aβ from enhanced γ-secretase activity increases stress levels generally. This is in agreement with a previous report that demonstrated that exogeneous addition of Aβ to cell culture media activates stress signalling, most likely through re-internalisation of aggregates [30]. The fact that stress levels in APP_S-I_ are slightly lower suggests that the differential processing of these two mutants may mitigate stress slightly. Aβ_42_ is known to aggregate more quickly than Aβ_40_, which could lead to the formation of larger oligomers in APP_S-I_ that are generally less toxic to cells [31]. In addition, the results suggest that different cell lines react to the application of environmental stress with different speeds. APP_S_ upregulates most markers earlier than APP_S-I_, which in turn reacts faster than APP_WT_. When extrapolated to patients, this could imply that a genetic predisposition in the form of a mutation in APP could result in a stronger signalling response to stress, leading to faster loss of neuronal cells.

### 2.2. Model-Based Comparison of Protein Load and UPR Kinetics for Wild-Type and Mutant APP Cell Lines Suggests Mutant APP Cell Lines Have a Higher Load of Unfolded Proteins and a Larger Transcriptional Response

To better understand the dynamics of the elaborate mammalian UPR system, we constructed a computational model integrating the response of three signalling branches; PERK, IRE1, and ATF6 (see Appendix A for model development). Parameters for IRE1 activation were extracted from the literature, while it was assumed that the activation of PERK and ATF6 followed the same kinetics. A detailed list of parameter values is presented in Appendix A. Parameters pertaining to the kinetic rates of UPR progression were estimated from the experimental data of the APP_S_ cells and are shown in Appendix A. These were fixed for the other two cell lines, with the exception of six parameters that were estimated separately from the data set for each cell line. The latter were the parameters representing the magnitude of the stress response for each cell line, $N_{ATF4_{m}}$, $N_{Bm},$
$N_{CHOP_{m}},\;N_{ATF6\alpha }\;,$ the folding rate $\gamma_{fold}$, and the unfolded protein load *K_u_* presented in Appendix A. This was necessary to reproduce the unique experimental behaviour of each cell line in terms of the upregulation profiles of the key UPR indicators, ATF4, BiP, and CHOP. APP_S_ exhibited a difference of 3.7- and 5.3-fold in the $N_{ATF4m}$ and $N_{CHOP_{m}}$ parameter values, respectively, when compared to APP_WT_, while APP_S-I_ rates were in the same order of magnitude as APP_WT_. We also observed changes in $N_{ATF6a}$, which were 76 and 57 times higher in APP_S_ and APP_S-I_, respectively. The changes in $K_{u}$ showed a 3.8-fold (APP_s_) and 1.4-fold (APP_S-I_) increase compared to APP_WT_. Overall, the fitted parameters were consistent with the experimental data that suggest that APP_s_ is the most stressed cell line, followed by APP_S-I_, followed by APP_WT_.

The model simulations using the fitted parameters accurately reflect the experimental results, as shown in Appendix A. APP_S_ showed a pronounced increase in the transcription of ATF4 $\left(N_{ATF4_{m}}\right)$ and the pro-apoptotic marker CHOP $(N_{CHOP_{m}}),$ as well as the magnitude of stress response following ATF6 activation $(N_{ATF6\alpha }$). Due to the lack of experimental data for the ATF6 pathway, we conducted a sensitivity analysis (Appendix A), which confirmed that increasing *N_ATF_*_6*α*_ results in the upregulation of XBP1T transcript levels and was used to fit the value that best reproduced the experimentally measured XBP1T levels with the current model assumptions. These simulation results are in accordance with a recent study by Yoshida et al. [26], which has illustrated that XBP1 transcription is induced by ATF6 activation under stress conditions and is spliced upon IRE1 activation. The simulation results for BiP $\left(N_{Bm}\right)$ also suggested that APP_WT_ cells are more resistant to stress, as they had a sharper transcriptional response of this gene and the lowest unfolded protein flux $K_{u}$ compared to APP_MUT_. The latter parameter, the value of which is indicative of the cumulative load of unfolded proteins in the ER, demonstrated a fold increase of 9.47 and 7.63 in APP_S_ and APP_S-I_, respectively, when compared to APP_WT_. Figure 5 shows the model-generated profile of the cumulative unfolded protein load for each cell line. Interestingly, the level of unfolded protein in APP_S-I_ at 3 h post-Tm treatment, when UPR indicators were significantly elevated compared to time zero levels, corresponds to that in APP_S_ at 2 h. This points to a potential threshold in unfolded protein load for UPR activation. The same value was reached at 4.5 h in APP_WT_, although the experimental system did not exhibit a significant response possibly because the lack of mutation means that unfolded proteins accumulating in the ER are easier to degrade through the ERAD pathway without stress activation.

### 2.3. The Computational Model Can Be Used as a Basis to Explore Treatments In Silico

Therapeutic strategies that modulate ER function represent a promising approach for prevention or treatment of neurodegeneration. Notably, it has been demonstrated that manipulation of the UPR can induce cellular adaptation and improvement of the disease phenotype [8,9]. Specific targets for ER regulation have been identified through screening of several compounds that could act as potential inhibitors of ER stress. These vary from inhibitors of the IRE1a pathway, such as the 4μ8C compound that blocks the splicing of XBP1 mRNA [32], inhibitors of PERK phosphorylation such as GSK2606414 [33], inhibitors of protein synthesis through enhancement of eIF2α phosphorylation, or inducers of ATF4 with salubrinal [34] and guanabenz molecules [35,36]. Another strategy involves increasing protein-folding capacity without triggering other stress mechanisms. Selective mood-altering drugs such as lithium [37,38], valproate, and very high (supratherapeutic) doses of carbamazepine are known to upregulate BiP expression without activating other stress markers [39]. We subjected the mathematical model to a series of case studies representing the biological effect of known therapeutics in order to demonstrate how the model can be used to explore treatment options in silico. The model for APP_S_ was chosen for these case studies since these cells underwent a stronger stress response in our experimental analysis.

#### 2.3.1. Case Study 1: Salubrinal Is Able to Quench Stress by Attenuating the Level of Unfolded Proteins

Salubrinal has been shown to inhibit eIF2α dephosphorylation and protect cells from stress by prolonging translational attenuation [34]. The computational model was modified to simulate this by removing the reaction of eIF2α dephosphorylation Appendix A. In other words, once an eIF2α molecule was phosphorylated, it would stay in that state. This has implications for Appendix A, in which the influx of new proteins to the ER is reduced with a decreasing number of non-phosphorylated eIF2α molecules. According to that equation, inhibition of eIF2α dephosphorylation is expected to shut down protein translation. Indeed, the model simulation results in Figure 6 demonstrate that salubrinal significantly reduces the load of unfolded proteins in cells. Consequently, the stress response is predicted to be less sharp compared to the response of the untreated cells, as is demonstrated to be the case. The level of upregulation of BiP mRNA levels due to unfolded protein accumulation is also lower in salubrinal-treated cells (Figure 6). These results are in agreement with the experimental observations of Sokka and co-workers [40], who demonstrated that salubrinal was able to counteract the effects of a chemical stressor in cultures of rat hippocampal neurons, as well as promoting survival in live animals. This suggests the model can accurately capture the biological effects of salubrinal.

#### 2.3.2. Case Study 2: The Reported Biological Effects of Lithium and Valproate Can Be Captured within Our Model System

Mood altering drugs and other compounds have been demonstrated to upregulate chaperone expression without inducing pro-apoptotic markers [39]. Hence, it has been proposed that they could be administered to patients pre-emptively to prevent the induction of the ER stress response and subsequent neuronal loss. To explore whether this would be feasible within our model system, we have performed a model-based case study to determine the level of BiP upregulation necessary to process all the unfolded proteins entering the ER without activating the UPR. In practice, the value of parameter $\beta_{Bm}$, which represents the transcription rate of BiP, was increased in the model for cells producing APP_S_ until no XBP1 mRNA splicing occurred. The simulation results presented in Figure 7 show that a nine-fold increase in BiP transcription would be sufficient for processing the unfolded proteins entering the ER for this cell line without inducing stress. This increase is within the capabilities of the system, which experimentally exhibited a 25-fold increase in BiP mRNA levels under conditions of stress.

.

## 3. Discussion

In the current study, we developed a framework for investigating ER stress through a combination of cell culture experiments and mathematical modelling that together can predict the effects of genetic predisposition and/or drug treatment on the ER stress response. In our experimental analysis, we investigated the transcript levels of a number of UPR target genes to create a comprehensive snapshot of the activity of all three signalling arms of the pathway. Our analysis showed that cells overexpressing APP695 bearing genetic mutations activated ER stress signalling differently and have transcriptional differences of the key regulators ATF4, BiP, and CHOP compared to those overexpressing wild-type APP695. This reinforces the idea that genetic background can affect the ability of cells to recover from environmental stress and could possibly contribute to the earlier onset of disease in vivo. Although our analysis concentrated on transcriptional response, faithful translation of the mRNA of the UPR target genes upregulated under induction of stress is essential, as part of the transcriptional UPR response [41,42] and measurements of transcriptional changes are more experimentally tractable for recalibrating the model for different genetic backgrounds in the future. Even though there is significant regulation of ATF4 on the translational level [43], we find that upregulation of ATF4 mRNA transcript level also occurs. Thus, it can serve as a more convenient proxy for stress activation via PERK signalling.

Using our experimental results, we established a computational model that explores the role of UPR in mammalian cells expressing FAD mutations and that was able to reproduce not only the desired shape of the response but also the relative increase for the five mRNAs whose upregulation was measured experimentally (BiP, ATF4, XBP1 total, XBP1 spliced, and CHOP). 

Given its accuracy in reproducing the experimental data, the model represents a resource to identify specific proteins in the stress response that could be targeted with therapeutics to stimulate different desired responses. As we have shown for salubrinal, the model accurately captures that it acts selectively on the PERK-mediated signalling pathway and maintains elevated eIF2α phosphorylation levels by inhibition of the phosphatases responsible for its dephosphorylation [35]. Similarly, for other, novel molecules, if the mechanism of action is known, the present model can be used to simulate the corresponding effect. Conversely, given a set of measurements of the ER stress response, the model can be used to infer the likely mechanism of action of a new drug whose biological mechanism is not known.

The switch to apoptosis in ER stress remains unclear; however, evidence shows that it is linked with the timing of the UPR. Lin et al. [44] showed that a tailored combination of individual UPR branches determines susceptibility to stress, which is in accordance with our data, which show that activation of the PERK pathway is sufficient to restore homeostasis in cells overexpressing wild-type APP695. This observation is in line with PERK’s posited role to overcome short-term stress. In contrast, APP_MUT_ had a stronger stress response that also required activation of the ATF6 and IRE1 pathways. The model suggests that the underlying molecular explanation for this is the higher unfolded protein load in APP_MUT_. In fact, the model predicts the load of unfolded proteins to be in the order APP_WT_ < APP_S-I_ < APP_S_, and the experimental data on stress levels corroborate this ranking. Nevertheless, in silico analysis illustrated that even in APP_S_, the levels of unfolded proteins returned to their initial values upon UPR activation, showing that UPR successfully resolves stress. 

This study presents the first mathematical model of the ER stress response in human cells that has been calibrated with in vitro experimental data, giving it a high degree of predictive power. It constitutes a first step towards personalised treatment planning for Alzheimer’s therapy and has potential for in silico screening of new drug compounds. Although this and other such models can be parameterised with relative ease using in vitro experiments, time-course in vivo measurements are unattainable. It is therefore difficult to confirm model applicability and generalisability in human patients. However, the establishment of an in vitro system using induced pluripotent stem cells (iPSCs) from AD patients [45] provides an ideal model system that allows for a variety of measurements that would otherwise be too invasive to perform. Comparison of measurements across experiments using iPSCs from large numbers of patients would also provide quantitative information on the natural variation in APP and UPR marker levels. Apart from being useful in parameterising the model proposed herein, such data would allow us to determine whether it is possible to build a generic computational model of FAD that can be used as a test bed for possible pharmacological interventions. Additionally, it would become possible to augment the present model with additional pathways known to be involved in disease progression, the most important of which is apoptosis. 

## 4. Materials and Methods

### 4.1. Cell Culture

SK-N-SH human neuroblastoma cells (ATCC^®^ HTB-11™) were maintained in Minimum Essential Medium Eagle (Sigma Aldrich, Gillingham, UK) supplemented with 10% foetal bovine serum (FBS) (Thermo Scientific, Loughborough, UK). Cells were grown in a monolayer at 37 °C in a humidified atmosphere with 5% CO_2_. Each culture was seeded in a new tissue culture flask at a density of 7.2 × 10^5^ cells/mL and passaged every 3 to 4 days. Viable cell concentration was determined by light microscopy using the trypan blue dye exclusion method.

### 4.2. Constructs of Mutant APP695

Generation of Point Mutants—Point mutants of the Swedish 670/671[Lys (AAG)-Met (ATG) to Asn (AAC)-Leu(CTG)] and Indiana 670/671 [Lys (AAG)-Met (ATG) to Asn (AAC)-Leu(CTG)] and 717 [Val (GTV) to Phe (TTC)] mutations were introduced into the APP695 coding sequence in pCI-neo mammalian expression vector (Promega, Hampshire, UK) by site-directed mutagenesis using the QuikChange^TM^ kit (Stratagene, Cambridge, UK) according to the manufacturer’s instructions. The fidelity of the mutated constructs was confirmed by sequencing (Eurofins MWG, Wolverhampton, UK). Oligonucleotide sequences used for mutagenesis are shown in the Appendix A.

### 4.3. Generation of Stable Cell Lines

SK-N-SH cells stably expressing APP695 containing either the wild type APP, the Swedish mutation alone, or APP with both the Swedish and the Indiana mutation were obtained by transfection using Lipofectamine 2000^TM^ (Thermo Scientific, Loughborough, UK) according to the manufacturer’s protocols. Forty-eight hours post transfection, colonies were selected using 500 µg/mL G418 (Sigma Aldrich, Gillingham, UK). The stable SK-N-SH cells overexpressing APP_WT_ and APP_MUT_ were maintained in the same medium as the non-transfected SK-N-SH cells supplemented with 50 µg/mL G418. At this stage, a master cell bank was created and all subsequent experiments were performed with a fresh vial of cells from this bank to minimise the likelihood of reduced expression.

### 4.4. SDS-PAGE and Western Blotting

Cells stored at −80 °C in a freeze mix containing 10% DMSO at a final concentration of 7.5 × 10^6^ were thawed. They were then centrifuged at 2000 rpm for 5 min, and pellets were harvested using the Mem-PER^®^ extraction kit (Thermo Scientific, Loughborough, UK). For analysis, concentrated NuPage sample buffer (4x) and 1M DTT were added to the samples, which were heated at 90 °C for 10 min. Equal amounts of protein were analysed on a gradient 4–20% precise protein gel (Thermo Scientific, Loughborough, UK) in Tris-Hepes running buffer at 120 V for 50 min. The proteins were transferred to a PVDF membrane (Milipore Sigma, Watford, UK) at 25 V for 1 h 30 min. The WesternBreeze^®^ Chemiluminescent Kit (Thermo Scientific, Loughborough, UK) was used for Western blotting. Membranes were incubated with rabbit monoclonal anti-APP at 1:200 dilution (Thermo Scientific, Loughborough, UK) and with mouse monoclonal anti-actin-β (Abcam, Cambridge, UK) as a loading control (1:500 dilution). Primary antibodies were detected using alkaline phosphatase-conjugated secondary antibodies, and bands visualized using a LAS-3000 luminescent image analyser (Fuji Photo Film, Tokyo, Japan).

### 4.5. Stress Induction Protocol

For all experiments, SK-N-SH cells that were ~80% confluent were used in order to rule out the influence of overgrowth on stress. Cultures were differentiated for 5 days using 10 µM all trans-retinoic acid (Sigma Aldrich, Gillingham, UK) supplemented to the culture medium. On the day of stimulation, the culture medium was renewed 1 h before the stress experiment to create uniform conditions. Cells were stimulated with 1 µg/mL tunicamycin [46] in culture medium containing retinoic acid.

### 4.6. MTS Assay

In vitro cytotoxicity of tunicamycin was determined with the MTS tetrazolium salt/phenazine methosulfate (PMS) solution (Promega, Hampshire, UK) according the manufacturer’s instructions. Briefly, differentiated cells in 6-well plates were exposed to tunicamycin (1 μg/mL) for 0 to 24 h. MTS solution (0.25 mg/mL) was added, and the cells were incubated for an additional 2 h at 37 °C, followed by absorbance measurements at 490 nm. 

### 4.7. RNA Isolation and cDNA Construction

Total RNA was isolated from actively growing cell cultures by harvesting them in 400 µL lysis reagent from the High Pure RNA isolation kit (Roche Applied Science, Penzberg, Germany), followed by spin column purification according to the manufacturer’s instructions. RNA integrity was electrophoretically verified by GelRed staining and by OD_260_/OD_280_ nm absorption ratio > 1.95 (Thermo Scientific, Loughborough, UK). The cDNA was prepared by quantitative real-time PCR (qRT-PCR) of total RNA (1 μg) using the Sensiscript^®^ reverse transcription kit (Qiagen, Manchester, UK). Oligonucleotide sequences used for PCR are shown in the Appendix A.

### 4.8. Optimisation of qRT-PCR Conditions

Conditions for all primer pairs were optimised by gradient PCR and analysed by gel electrophoresis to find conditions that produced only the target amplicon. Primer efficiencies were determined by performing serial dilutions of cDNA two-fold up to 6 points followed by qRT-PCR, which was performed in duplicate using 4 different cDNA concentrations. Efficiency calculation was performed according to Pfaffl et al., and primers were all optimised to an equal annealing temperature of 64 °C [47].

### 4.9. Quantitative Reverse Transcription Polymerase Chain Reaction (qRT-PCR)

Individual qRT-PCR reactions were performed using a mastermix containing SYBR^®^ Green JumpStart™ Taq ReadyMix™ (Sigma Aldrich, Gillingham, UK), 100 ng cDNA, and a final primer concentration of 1600 nM. Each reaction was performed in triplicate in 96-well plates (Eppendorf, Stevenage, UK) sealed with thermal film (Bioline, London, UK) and run in Mastercycler^®^ ep Realplex 4S device (Eppendorf, Stevenage, UK). The conditions for the two-step PCR were as follows: single denaturation cycle at 92 °C for 2 min followed by 40 cycles of 92 °C for 30 s, 64 °C for 30 s, and a final extension cycle of 72 °C extension for 1 min. Amplified products were validated by melting curve analysis (62.7 °C to 92 °C with 0.025 °C per second increment) and agarose gel electrophoresis. C_t_ values were determined by the Mastercycler ep Realplex software with a threshold of 100 (arbitrary fluorescence units). For each cell line, ER stress marker (e.g., BiP, CHOP) mRNA levels were normalised for the expression of the endogenous housekeeping gene 18S rRNA. A control lacking reverse transcriptase was used to assess the level of contamination of genomic DNA. Samples with Ct values < 40 in this control were discarded and reprepared.

### 4.10. Statistical Analysis

Statistical analysis was conducted using SPSS software 2013 (Statistical Package for Social Sciences SPSS Inc., Chicago, IL, USA). Evaluation of the differences among treatments between and within the cell cultures was performed using analysis of variance (one-way ANOVA). Levene’s test was applied to compare the variances among treatments and determine their homogeneity. When the variances were statistically homogeneous (*p* < 0.05), ANOVA was employed, and when they presented heterogeneity (*p* > 0.05), the Welch ANOVA procedure was applied. In the case that the aforementioned tests indicated significant effects (*p* < 0.05), the data were subjected to Tukey’s honestly significant difference (HSD) and Games-Howel respectively for post hoc analysis to assess whether the means of each treatment were significantly different. Significance was considered at * *p* < 0.05, ** *p* < 0.01 and *** *p* < 0.001, as indicated in each case.

## 5. Conclusions

We showed that FAD APP mutations trigger UPR activation more quickly than the cells overexpressing wild-type APP. The cells overexpressing APP_S_ were the most sensitive to ER stress. Our computational model reflects the biological reality of ER stress activation but, as is the case with most models of biological systems, it is based on simplifying assumptions. A higher fidelity of the mammalian UPR model would be achieved by estimating protein-level parameters using data from high throughput proteomic studies if they become available. Furthermore, coupling the model with an individualised wet lab system could bring the development of personalised treatments for AD and other neurodegenerative diseases one step closer. Such an experimental system has recently been established for iPSCs from patients [45] and would allow a more accurate determination of patient-specific parameters and, thus, tailored treatment avenues.

## Figures and Tables

**Figure 1 ijms-22-12001-f001:**
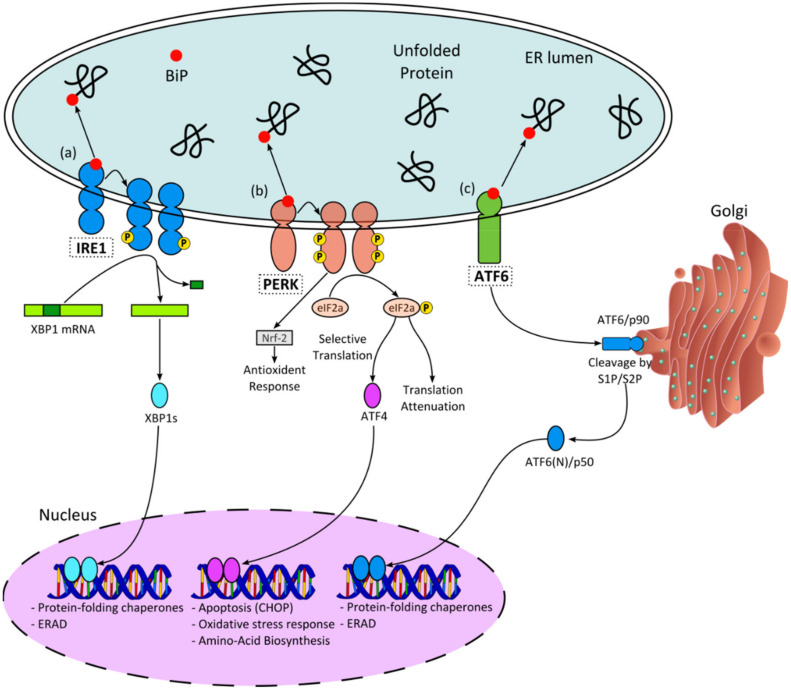
Mammalian UPR pathway: Upon ER stress, BiP dissociates from the ER membrane sensors IRE1, PERK, and ATF6, allowing activation of their downstream signalling cascades, including the unfolded protein response (UPR). (**a**) IRE1 dimerisation and autophosphorylation result in activation of an endoribonuclease activity that removes an intron in the mRNA encoding XBP1 transcription factor (TF). This reaction is essential for the translation of the active XBP1s TF, which induces UPR (i.e., chaperones) and non-UPR (ERAD) target genes. (**b**) PERK phosphorylates eIF2α (p-eIF2α), leading to decreased protein synthesis (translation attenuation). Paradoxically, lower levels of eIF2α promote translation of ATF4, a selective TF that drives expression of UPR target genes (e.g., CHOP). (**c**) ATF6 translocates from the ER to the Golgi apparatus, where it undergoes cleavage and results in the release of the active p50 fragment in the cytosol. The p50 translocates to the nucleus, where it binds to and activates the ERSE consensus sequence CCAAT-N9-CCACG found in the promoter of several UPR-target genes, including BiP, CCAAT/enhancer-binding protein (C/EBP) homologous protein (CHOP), and XBP1.

**Figure 2 ijms-22-12001-f002:**
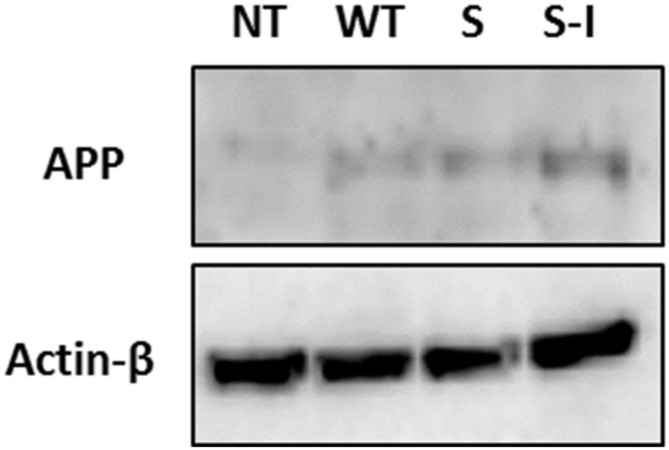
Establishing a stably transfected FAD cell-model system. A representative Western blot (WB) indicating APP overexpression in stably transfected SK-N-SH cells. Expression of APP695 in cells overexpressing wild type (APP_WT_), Swedish (APP_S_), or Swedish-Indiana (APP_S-I_) compared to the non-transfected cells (NT). Actin-b was used as a loading control.

**Figure 3 ijms-22-12001-f003:**
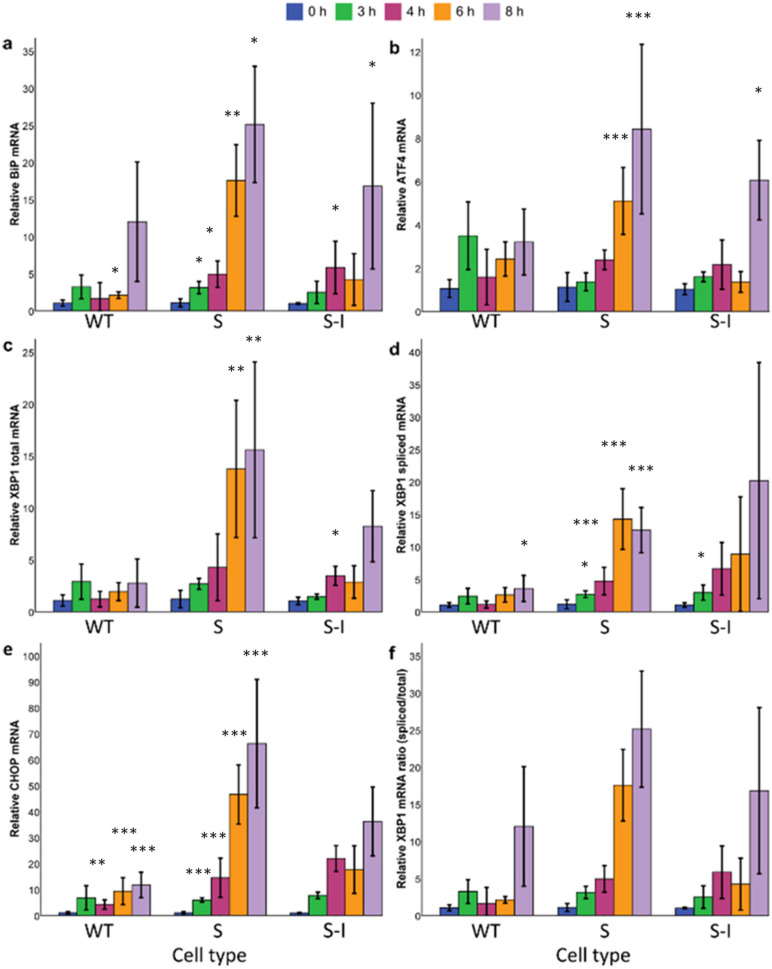
qRT-PCR analysis to monitor the expression of UPR markers in a cell-model system of FAD during Tm treatment. Differentiated SK-N-SH cells stably overexpressing wild-type (APP_WT_), Swedish (APP_S_), or Swedish-Indiana (APP_S-I_) mutant forms of APP695 were treated with tunicamycin (Tm) for 8 h. The transcript levels of the UPR targets (**a**) BiP, (**b**) ATF4, (**c**) XBP1 total, (**d**) XBP1 spliced, and (**e**) CHOP were assessed with qRT-PCR. Samples were normalized to 18S rRNA and the 0 h time point for the respective gene. (**f**) The ratio of XBP1 mRNA spliced/total was calculated from the qRT-PCR data. Values represent the mean for n = 5. These are from two independent biological experiments, where the technical replicates for each experiment were n = 3 and n = 2, respectively. Each sample was analysed with n = 3 qRT-PCR technical replicates. * *p* < 0.05, ** *p* < 0.01, *** *p* < 0.001.

**Figure 4 ijms-22-12001-f004:**
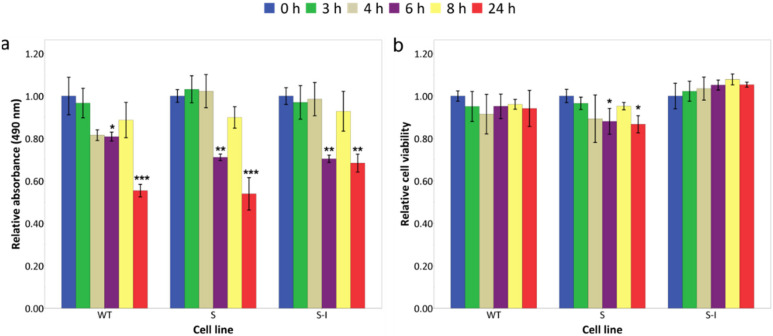
Effect of Tm on proliferation and cell viability in a FAD cell-model system. Cell proliferation and viability of differentiated stressed cells were analysed by (**a**) MTS assay and (**b**) trypan blue dye exclusion method following the incubation of cells with Tm for the indicated time points. Values represent the means for (**a**) n = 3; ± SD and independent of (**b**) representing two independent experiments where the technical replicates for each experiment were n = 3 and n = 2, respectively. For 24 h the means are from n = 3. The results are presented in relation to the zero-hour Tm treatment. Statistically significant differences demonstrated on the graph are from cells treated with Tm for zero hours compared with the cells treated with Tm for 3 h, 4 h, 6 h, 8 h, or 24 h, respectively. * *p* < 0.05, ** *p* < 0.01, *** *p* < 0.001.

**Figure 5 ijms-22-12001-f005:**
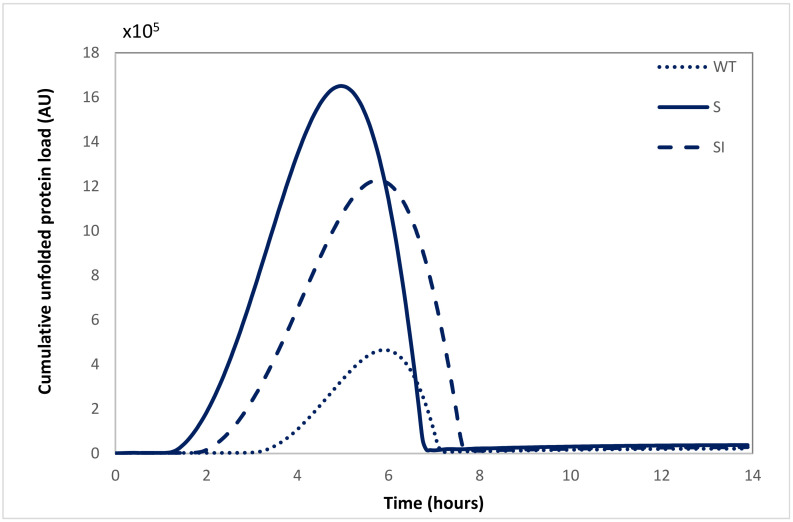
Model-generated profile of the unfolded protein load in each cell line with respect to time post-Tm treatment.

**Figure 6 ijms-22-12001-f006:**
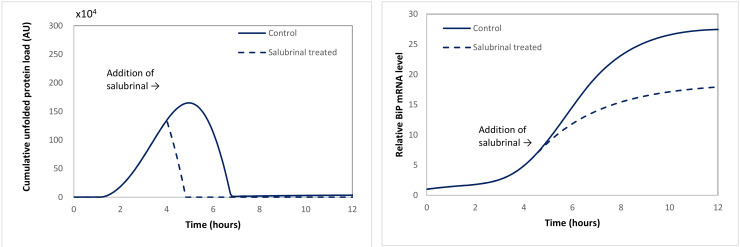
Effect of salubrinal on the cumulative unfolded protein load (**left**) and the BiP mRNA level (**right**) in APP_S_ cells during Tm treatment.

**Figure 7 ijms-22-12001-f007:**
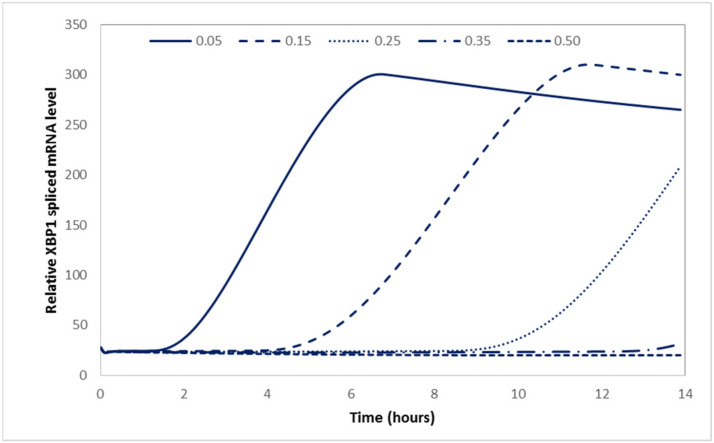
Effect of valproate in APP_S_ stressed cells during Tm treatment. Simulation results for APP_S_ show the effect of VPA on spliced XBP1 mRNA levels by means of varying BiP transcriptional rates $\left(\beta_{B_{m}}\right)$.

**Table 1 ijms-22-12001-t001:** Viable cell concentration normalised against initial value for n = 3. Student t-test analysis revealed a significant difference between the 0 h and 24 h stress induction in all three cell lines. *p* < 0.05 for APP_WT_ and APP_S_ and *p* < 0.01 for APP_S-I_.

TM-Treatment		Normalised Viable Cell Concentration		
		WT	S	S-I
0 h	Mean	1.00	1.00	1.00
	SD	1.23 × 10^−1^	1.80 × 10^−1^	8.21 × 10^−2^
24 h	Mean	6.52 × 10^−1^	5.83 × 10^−1^	5.30 × 10^−1^
	SD	6.91 × 10^−2^	1.06 × 10^−1^	1.14 × 10^−1^

## Data Availability

Experimental data and mathematical models are available upon request.

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
