# Peer review of "Model Identifies Genetic Predisposition of Alzheimer’s Disease as Key Decider in Cell Susceptibility to Stress"

_ijms, 2021, doi:10.3390/ijms222112001_

Round 1

Reviewer 1 Report

Overall, the manuscript is interesting and informative. The paper raises a very interesting point that mutations in APP associated with FAD lead to different magnitude of ER stress response signaling. Model proposed by the Authors shows the load of unfolded proteins to be in the order APPWT < APPS-I < APPS and the experimental data on stress levels corroborate this ranking.

Nevertheless, I have several questions to be answered. For this reason, I recommend minor revision of the paper.

Comments and Suggestions for the Authors:

  1. The abstract should be more detailed / more informative.
  2. Do the Authors think that their model using primary cultures / iPSCs would be better? It would be appropriate to discuss this issue.
  3. How the stability of the construct was verified.
  4. The whole images of wb should be added as a supplementary data.
  5. There is no information about replicates and n number in the Fig. 2.
  6. Does the cell culture was checked against mycoplasma contamination?
  7. The loading control anti-actin B dilution (1:500) is quite high and can thus eliminate the differences in the obtained results. I would recommend dilution above 1:2000.
  8. Please provide dilution of secondary antibodies.
  9. Part 5.5. – some special characters / prefixes are missing.
  10. Part 5.9. – how the authors verified the contamination of DNA genomic samples?

Author Response

We thank the reviewer for their invaluable feedback and suggestions. Below is a point-by-point response to specific comments.

  1. The abstract should be more detailed / more informative.

We revised the abstract to provide more detail on the work described and results obtained in this study.

  1. Do the Authors think that their model using primary cultures / iPSCs would be better? It would be appropriate to discuss this issue.

We agree with the reviewer that data from iPSCs would be more informative than the model system used in the paper and could lead to the development of personalised treatments. Indeed, a comment to that effect was already included in the conclusions section of the manuscript, but we have included additional thoughts in the last paragraph of the discussion on page 10.

  1. How the stability of the construct was verified.

This is not something that we investigated because every experiment was conducted with a fresh vial of cells from the original cell bank. This strategy ensured minimal accumulation of generation numbers, which should also have minimised the probability of silenced or reduced expression. We have added a sentence explaining this strategy to section 5.3.

  1. The whole images of wb should be added as a supplementary data.

We have provided the WB images in the revised version of the manuscript.

  1. There is no information about replicates and n number in the Fig. 2.

This is a representative Western blot. We have updated the figure reference to clarify this.

  1. Does the cell culture was checked against mycoplasma contamination?

Yes, both the parental and all mutant cell lines have been checked for mycoplasma contamination.

  1. The loading control anti-actin B dilution (1:500) is quite high and can thus eliminate the differences in the obtained results. I would recommend dilution above 1:2000.

We thank the reviewer for their recommendation. Unfortunately, we have no samples left from this experiment to repeat this analysis with the recommended dilution.

  1. Please provide dilution of secondary antibodies.

We used the Western Breeze detection kit where the secondary antibodies are provided as a solution and used it according to the manufacturer’s instructions. 

  1. Part 5.5. – some special characters / prefixes are missing.

Thank you for spotting this mistake. We have corrected the units.

  1. Part 5.9. – how the authors verified the contamination of DNA genomic samples?

A control lacking reverse transcriptase was used to assess the level of contamination of genomic DNA.  Samples with Ct values <40 in this control were discarded and reprepared. We have added this detail to section 5.9.

Reviewer 2 Report

This is an innovative, well-designed and well-performed study, which reports that developed by authors new strategy could be a useful tool in finding a powerful therapeutic strategy. The authors by creating the cell lines that mimic a genetic predisposition to Alzheimer’s, causing them to environmental stress, and using the resulting experimental data to formulate a mathematical model aimed to e.g. elucidate the short-term response to stress. Computational models appear to be one of the most informative and valuable tools in predicting some experimental observations, which in the next step should be confirmed.  

Minor point.

An Introduction - 
I suggest described detailed UPR response eg. which factors may influence on upregulation of the response, if the level of the response depends on age. 

It seems to be interesting write short paragraph about the limitation of this kind of studies (computational models in predicting the biological mechanisms and safety of gene therapy in long-term administration). 

Author Response

We thank the reviewer for their invaluable feedback and suggestions. Below is a point-by-point response to specific comments.

An Introduction - 
I suggest described detailed UPR response eg. which factors may influence on upregulation of the response, if the level of the response depends on age. 

Due to lack of in vivo data and differences among AD patient pathophysiology, we cannot speculate on the possibility of age-depended response or APP levels. We have amended the last paragraph of the introduction (page 3) to explain that these mutations are linked to genetic predisposition to AD, also known as familial AD, which is responsible for most cases of early disease onset.  

It seems to be interesting write short paragraph about the limitation of this kind of studies (computational models in predicting the biological mechanisms and safety of gene therapy in long-term administration). 

 We thank the reviewer for this suggestion. We have added a paragraph with our thoughts on the limitations of computational models and their parameterisation using in vitro model systems, as well as some suggestions on promising alternatives in the last paragraph of the discussion on page 10.